# Knowledge Graph-Based Framework to Support Human-Centered Collaborative Manufacturing in Industry 5.0

László Nagy , János Abonyi  and Tamás Ruppert *

HUN-REN-PE Complex Systems Monitoring Research Group, Department of Process Engineering, University of Pannonia, 8200 Veszprém, Hungary; nagy.laszlo.92@gmail.com (L.N.); janos@abonyilab.com (J.A.)
* Correspondence: ruppert@abonyilab.com

**Abstract:** The importance of highly monitored and analyzed processes, linked by information systems such as knowledge graphs, is growing. In addition, the integration of operators has become urgent due to their high costs and from a social point of view. An appropriate framework for implementing the Industry 5.0 approach requires effective data exchange in a highly complex manufacturing network to utilize resources and information. Furthermore, the continuous development of collaboration between human and machine actors is fundamental for industrial cyber-physical systems, as the workforce is one of the most agile and flexible manufacturing resources. This paper introduces the human-centric knowledge graph framework by adapting ontologies and standards to model the operator-related factors such as monitoring movements, working conditions, or collaborating with robots. It also presents graph-based data querying, visualization, and analysis through an industrial case study. The main contribution of this work is a knowledge graph-based framework that focuses on the work performed by the operator, including the evaluation of movements, collaboration with machines, ergonomics, and other conditions. In addition, the use of the framework is demonstrated in a complex use case based on an assembly line, with examples of resource allocation and comprehensive support in terms of the collaboration aspect between shop-floor workers.

**Keywords:** human-centered; knowledge graph; Industry 5.0; manufacturing ontology; semantic reasoning; operator support

## 1. Introduction

The global economy and the developers of MES (Manufacturing Execution System) and ERP (Enterprise Resource Planning) systems face the challenge of enhancing productivity while retaining human labor in the manufacturing sector [1]. With the increasing diversity and complexity of product lifecycle applications, there is a growing need to digitize knowledge related to various aspects of the industry, such as process planning, production, and design. It is suggested that the key drivers for transforming data into knowledge and advancing process automation through interoperable data will involve KGs (knowledge graphs), semantic web technologies, and multi-agent systems [2]. Effective representation and communication of domain knowledge are also vital for smart manufacturing. The primary focus of Industry 4.0 lies in achieving extensive digitization, while Industry 5.0 aims to merge cutting-edge technologies with human involvement, characterized as a value-driven approach rather than a technology-centred approach [3]. Industry 4.0 integrates digital technologies such as IoT, artificial intelligence, big data, and automation into manufacturing processes to improve their efficiency, productivity, and customization. By prioritizing connectivity, data exchange, and smart factories, it establishes a more adaptable and responsive manufacturing setting. Industry 5.0, an emerging concept based on Industry 4.0, reintroduces human-centric methods into manufacturing operations. In contrast to Industry 4.0, which focuses mainly on automation and machine-to-machine communication, Industry 5.0 emphasizes collaboration between humans and machines.

The objective is to take advantage of the distinct strengths of humans and machines, such as creativity, problem-solving, and emotional intelligence, to drive increased levels of innovation, flexibility, and sustainability in manufacturing. Industry 5.0 strives to achieve a harmonious balance between technology and humanity, using advanced technologies while giving priority to human well-being, creativity, and empowerment. Furthermore, there is growing interest in research areas such as industrial humanization [4], sustainability, and resilience [5]. Within the context of Industry 5.0, the importance of a Knowledge Graph (KG) is underscored by its ability to represent and analyze intricate data related to human operators [6].

The networked data structure of this system effectively records essential operator-related elements like ergonomics, working conditions, and machine cooperation in a systematic manner. By utilizing ontologies, a structured understanding of human actions in the manufacturing setting is achieved, guaranteeing compliance with human limitations and abilities. The graph's dynamic query and visualization features support real-time monitoring and flexibility. Consequently, the Knowledge Graph (KG) has become a key instrument for enhancing human-centered approaches in intricate manufacturing environments [7].

The human-centered aspect of the Industry 5.0 idea [3] strives for improved human–machine interaction, envisioning robots integrated with the human mind to collaborate rather than compete [8]. Throughout history, humans have influenced cyberphysical systems (CPSs) significantly, playing a vital role in their establishment and advancement. Consequently, human intelligence stands out as a crucial and predominant element in intelligent manufacturing, aligning with the concept of human-cyber-physical systems (H-CPSs) [9]. To achieve a suitable level of human–machine fusion, the concept of operator 4.0 [10,11] must be assessed. This concept promotes adaptive automation within collaborative human-automation work systems, fostering a socially sustainable manufacturing workforce. A more recent idea, the concept Resilient Operator 5.0 [12], explores improving the resilience of human operators to various workplace factors, thus facilitating the implementation of efficient smart manufacturing systems. Additionally, a proposition is made to model cognitive abilities and task requirements using a human asset administration shell [13]. Ontology models can also help contextualize key performance indicators (KPIs) [14], identify indirect effects or influences, and analyze relationships within a complex network [15]. They can also assist in visually representing KPI themes, developing dashboards [16], and consolidating KPI-related data [17]. Once this relationship with the decision variable is established, it enables responsive development and optimization. The ontologies, semantic tools, and industry standards proposed in this paper can support the development of systems that enhance operators' resilience, flexibility, and efficiency. The primary contribution of this study is the introduction of a framework known as the Human-Centric Knowledge Graph (HCKG), which models elements related to the human operator, such as monitoring movement, work environment, and collaboration with robots, using ontology and standards. The framework is exemplified through an industrial case study and incorporates graph-based data querying, visualization, and analysis. An instance involving a complex wire harness assembly process illustrates instances of resource allocation and comprehensive support for human–machine collaboration. The key innovations and contributions of this paper include the following:

- Suggested the expansion of automation standards like ISA-95, AutomationML, or B2MML to cover human-centric processes and the application of semantic technologies.
- Advocated for a Knowledge Graph (KG)-based approach to bolster human-centered and collaborative manufacturing in Industry 5.0.
- Showcased a replicable industrial case study to validate the concept. Various graph-based analyses utilizing normal, directed, or hypergraphs will be demonstrated, such as resource allocation assessment, KPI evaluation, or identification of diverse collaboration forms between human and machine agents in the assembly process.

This research builds upon a previous conference paper [18], which introduced only a portion of the concept. Initially, Section 2 presents the current state of the field, pinpointing

knowledge gaps and motivations. The core contribution of the paper is outlined in Section 3, where the foundational elements of the HCKG design concept are elaborated. Section 4 outlines a wire harness assembly-based case study to trial the human-centered KG-based design concept. Lastly, the contributions and potential avenues for future research are discussed in Section 5.

## 2. State-of-the-Art—Knowledge Gap and Motivation

The integration of collaborative robots into manufacturing processes, known as human–robot collaboration (HRC), represents a significant advancement in Industry 4.0. Unlike traditional industrial robots that are confined to isolated cells, collaborative robots are designed to work alongside humans, using embedded interaction, sensing, and safety technologies. This enables a hybrid production environment where human and robot resources are dynamically allocated to optimize productivity, flexibility, and reconfigurability. HRC environments aim to overcome the limitations of manual and robotic assembly lines by providing a novel approach to task allocation and execution that improves overall manufacturing efficiency and adaptability [19].

Cyber-Physical Production Systems (CPPS) integrate physical processes with digital technologies to optimize production processes. These systems enable seamless communication between physical components and digital systems, fostering an adaptive and intelligent manufacturing environment that is capable of self-optimization and autonomous decision-making. Information management of emerging industry trends requires an effective solution, such as KGs, that uses a graph-based data model to capture knowledge in application scenarios that involve integrating, managing, and extracting value from diverse data sources, even on a large scale [20]. Semantic technologies such as ontologies, graph databases, semantic analytics, and reasoning provide an efficient way to process large amounts of data from multiple sources by making the entire data set transparent and accessible [21,22]. Semantic networks and graph-based analytics are recommended to handle process information using linked data features. Knowledge graph (KG) techniques are capable of extracting data from structured, semi-structured, or unstructured sources and then incorporating this information into a graph-based knowledge representation [23]. To improve operator working conditions, various monitoring systems, such as sensor networks, can be utilized to monitor operator movements and physical states, enabling the assessment of performance metrics [24,25]. In the process of ontology engineering, systematically adapting ontologies to different production systems and factory settings requires a thorough initial requirements analysis to ensure deep understanding [26]. Existing ontologies are meticulously assessed to facilitate reuse and adaptation through a modular design. Emphasizing granularity and maintaining consistent naming conventions requires thorough documentation. Seamless integration with existing factory data sources is carefully coordinated, and an iterative refinement strategy, supported by expert input and real-world tests, is integrated [1]. Utilizing specialized software and tools, along with adherence to standards like the OWL (Web Ontology Language) and the RDF (Resource Description Framework), coupled with stringent governance, guarantees compatibility and systematic updates. A knowledge reasoning framework has been suggested, using semantic data to improve real-time data processing in a smart factory setting [27]. A machine learning semantic layer has been introduced to complement augmented reality solutions in the industry, providing an intelligent layer [28]. Operators in an Industry 5.0 environment should be able to interact effortlessly with industrial assets while dealing with more complex assets. To achieve this development objective, a generic semantics-based task-oriented dialogue system framework like KIDE4I (Knowledge-drIven Dialogue framEwork for Industry) can offer a solution to reduce cognitive load [29].

The use of international standards can improve the quality of information systems by facilitating the interoperability of the software tools used. ISA-95 [30] is one of the essential standards in the field of integration of enterprise control systems and serves as a widely used basis for designing Industry 4.0 [31], IIoT (Industrial Internet of Things) [1] or smart

factories [32] related to MES and MOM (Manufacturing Operations Management). To create a semantically integrated design concept, the production capability and personnel models of the ISA-95 standard are recommended as a basis for modeling.

B2MML is an implementation of IEC/ISO 62264 [33] to provide a freely available XML for manufacturing companies [34]. B2MML standard elements are recommended for the development of problem-specific ontologies, such as the concept of collaborative assembly workstations [35], where semantic technologies are used to improve interoperability with external legacy systems such as ERP and MES. AutomationML [36] aims to standardize the exchange of data in the engineering process of production systems. In an AutomationML environment, the IEC 62264-2 personnel model [37] provides a method to model the operator in a production process with the following elements: personnel class, personnel class property, person and person property. AutomationML is also recommended as an exchange file format as a step toward automated job design based on optimized resource allocation [38].

Another important point to consider is that human-centric Cyber-Physical Production Systems (CPPSs) in smart factories and active collaboration between humans and machines have introduced an ontological framework known as the PSP ontology (Problem, Solution, Problem-Solver Ontology) [39]. The research focused on integrating the three main concepts of "Problem-Solving Semantically Profile", "Problem-Solver Profile", and "Solution Profile". In addition to the semantic representation and reasoning of these core concepts, the study introduced the contingency vector, competence and autonomy vectors, and the solution maturity index for CPPS [39]. Moreover, due to the insufficient operator-based models, particularly in decision-making aspects [40], it is recommended to incorporate the human operator model into the shop floor control system. Enhancing human–machine interaction through ontologies is recommended. To prioritize human well-being while ensuring production efficiency, the development of a human-centred intelligent environment requires the consideration of various factors. There is a high demand for identifying suitable factors to evaluate human–robot collaboration and ergonomic conditions for factory workers [41]. A comprehensive framework is required to evaluate Human–Machine Interfaces (HMI) and Human–Robot Interactions (HRI) in collaborative manufacturing settings [42]. A comprehensive systematic review [15] categorized the measures, indicators, and quality factors used in the HRI literature using a methodical approach. The indicators are grouped into categories relevant to industry 5.0 research, including physical ergonomics (safety, physical workload, job design), cognitive ergonomics (mental workload, awareness), performance (efficiency, effectiveness), and user experience satisfaction/hedonomics (emotional responses, acceptance, attitudes, trust) [15]. In the field of ontology engineering, especially within the Industry 4.0 framework, a structured series of methodical procedures is adhered to [43]. Initially, a comprehensive analysis of requirements is performed to grasp the diverse points of view of stakeholders and establish the goals, extent, and requirements of the ontologies [44]. Before embarking on any new development, existing ontological resources are evaluated for potential repurposing, focusing on efficiency and the integration of well-established concepts. Throughout the development phase, specialized tools are utilized to define essential elements such as classes, relationships, and axioms. A thorough assessment process, involving expert evaluations and automated validations, ensures alignment with recognized standards and specifications. Elaborate documentation is generated that elucidates the structure and operation of the ontology to enhance comprehension and stakeholder participation. The final result is incorporated into the Industry 4.0 setting and, acknowledging the evolving nature of industrial domains, undergoes regular reviews and improvements to maintain relevance and efficiency [45].

In summary, the knowledge gaps in a semantic framework to support collaborative and ergonomic manufacturing include the need for effective human-centered design integration [46], comprehensive manufacturing ontologies [47], robust semantic reasoning techniques [48], and advanced operator support tools [49]. Addressing these gaps requires overcoming challenges related to data integration, data quality, real-time analytics, and scalability in KG-based frameworks [50]. The motivation of this paper is to propose a

semantic-based framework for human-centric manufacturing and to present an industry-related case study of KG utilization. In the following section, the concept of HCKG design is presented after discussing the motivation for this research.

## 3. Human-Centered Knowledge Graph-Based Concept towards Collaboration in Manufacturing

This section delves into the primary contribution of this paper, which is the design concept of the Human-Centered Knowledge Graph (HCKG). Section 3.1 explores the activity model associated with the management of manufacturing operations. Section 3.2 describes the different human–robot collaboration scenarios and the key essential performance indicators of a human-centric assembly process. Lastly, Section 3.3 details the framework of the HCKG concept. The objective of the HCKG design concept is to establish a framework to monitor and control human–machine collaboration, improve resilience and agility, and improve working conditions for operators. The knowledge graph incorporates monitored data concerning the operator's activities, the environment, as well as all robots and equipment within the manufacturing space. Through the analysis of the collected knowledge graph data, collaboration can be enhanced, work instructions can be customized for the operator, and any modifications can be adaptively managed. Figure 1 illustrates the integration approach of the HCKG concept. In the initial segment, the Production Process element represents the intricate production environment encompassing all human–machine resources, processes, activities, and interactions. The Monitoring System element interacts with the production process, collecting historical and real-time data using sensors and IoT devices. Furthermore, the schema element offers semantic tools to establish a contextualized data model, while the meta-information element contains meta-information, such as industry standards, to ensure reusability. In knowledge graphs, Metal pertains to data about the data itself, providing details such as its source or properties, whereas Scheme defines the structure, properties, and relationships within the graph, organizing and standardizing the data model. The initial segment comprises a variety of structured and unstructured data sources that require preprocessing.

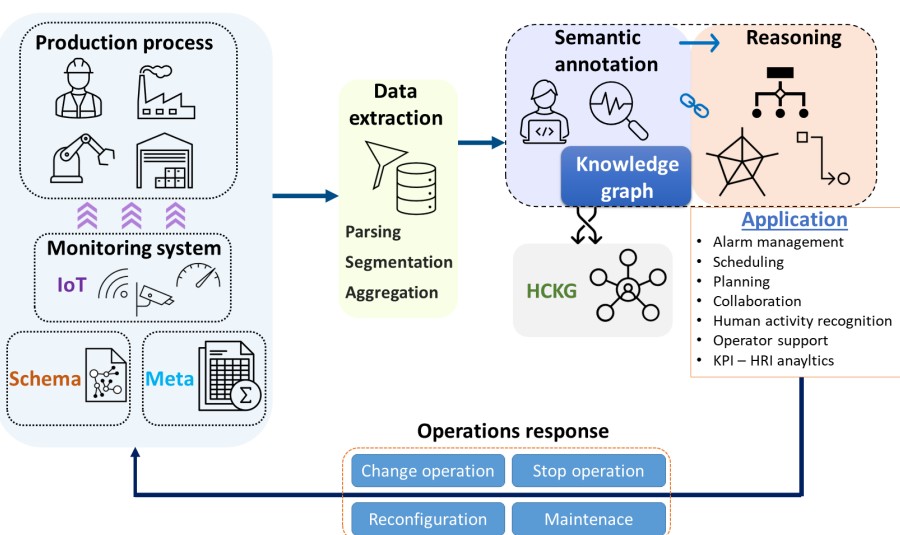

**Figure 1.** Integration of the HCKG design concept connected to the production process, using five segments.

Therefore, the second part involves procedures such as parsing, segmenting, and consolidating data, which constitute the Data Extraction component. The goal of data extraction is to recognize and retrieve pertinent information from unstructured or semi-structured data sources of the initial part and to transform it into a structured form for analysis and optimization. The third section encompasses the Semantic Annotation and Reasoning components, which leverage semantic modeling and data analysis in a complex

KG. The Semantic Annotation module constructs the KG utilizing the schema, metadata, and extracted data. This process entails the appending of metadata, standardized labels, or tags to entities and relationships within the KG, such as industry-specific terms or concepts from a particular domain. Within knowledge graphs, "metadata" refers to semantic details that delineate the attributes, context, and connections of the data, thereby enhancing comprehension and interpretation of the underlying information. It enables applications to recognize and classify diverse entities more precisely within the KG by furnishing additional context and facilitating more accurate categorization and identification within the data model. The HCKG module signifies the human-centric KG aspect of the established semantic network, which may encompass the entire KG or only the portion relevant to shop floor workers, depending on the specific scenario. This section is elaborated further in Appendix A and the case study in Section 4.2. The Reasoning component enhances the semantic information for the subsequent segment, the component Application . The reasoning process relies on the concept that the interrelations and links among various entities in the KG can be leveraged to derive logical inferences and form novel predictions. In the realm of analytics and optimization, semantic reasoning plays a crucial role in uncovering patterns, correlations, and causal connections between entities within the KG. Through the application of semantic reasoning, patterns and correlations among various data points can be identified, such as determining which machines are likely to cause delays on a specific production line. By engaging in reasoning across the KG, the application can determine the most efficient sequence of steps in the production process to minimize waste and improve efficiency. Human-centric KG applications can assist in managing alarms, scheduling operations or personnel, optimizing human–machine interactions, recognizing human activities, or analyzing performance metrics. This topic is further elaborated in Section 4.3 through a case study. The outcome of the application, along with the analysis results, leads to the operation response (fifth segment), which is then directed to the production process component. Examples of responses include change operation , stop operation , reconfiguration , or maintenance .

### 3.1. Manufacturing Operations Management

This section delves into an expanded MOM activity model, illustrated in Figure 2, where the components can be categorized based on the timing of their occurrence during the execution of the task. Although this aspect of the methodology has been previously documented by the authors in a conference paper [18], the inclusion of the associated MOMs here aims to enhance the comprehension of the HCKG framework. The temporal perspective of the general activity model concerning pre-, during, post-, and reference data is also emphasized [51]. Moreover, the supplementary modules for extending the standard activity model of MOM [52] are depicted in brown below.

The MOM approach is designed to provide a detailed insight into the mechanisms linked to the operator's role in a general manufacturing task, while also emphasizing the characteristics of the additional monitoring and support framework components. The generic activity model is segmented into four sections based on a temporal perspective, indicated by green labels in the diagram, and is evaluated and deliberated upon in a similar manner. The Reference Data encompasses all the details regarding individual operators, including their skills, capabilities, and expertise in specific domains. The Resource and Definition Management segments of the MOM system compile this data and establish the foundational information for the subsequent operational segments of the model. As an expansion of the reference data segment, the Control and Optimization component is suggested, where the integration of machine learning or artificial intelligence solutions can enhance the ongoing manufacturing processes.

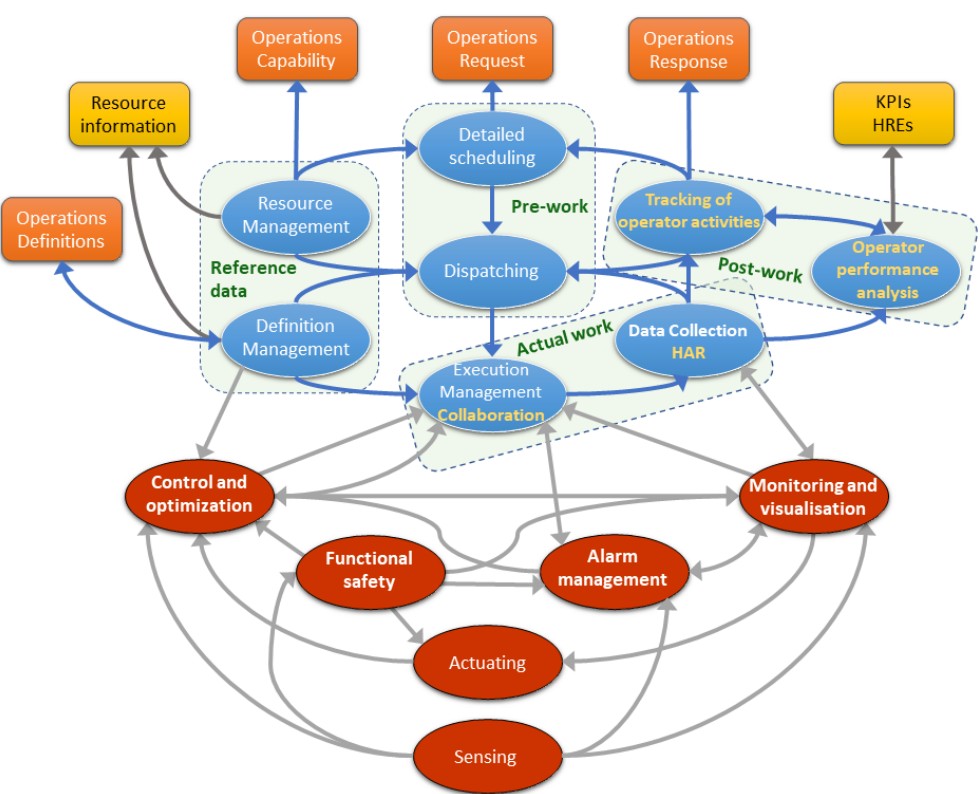

**Figure 2.** Activity model of manufacturing operations management from an operator-centric point of view [18].

The following section in Figure 2 represents the Pre-work phase, where Detailed Scheduling is employed based on the Operation Request, and Dispatching tasks are carried out. These actions ensure that all operators receive proper instructions, are efficiently scheduled, and are assigned tasks accordingly. The actual work segment of the MOM delineates the ongoing activities managed by Execution Management, while concurrently conducting Data Collection. Some elements centered on humans are integrated (marked in yellow text), such as Collaboration or the utilization of Human Activity Recognition (HAR) sensor technologies. To enhance real-time operator assistance, additional components like Alarm management, Monitoring and visualization are included as supplementary features. An intelligent monitoring system can gather data from various manufacturing parameters such as temperature, noise, or vibration, and display it graphically in real-time, issuing alerts in case of anomalies.

In the post-work phase of the task, monitoring of operator activities is carried out to derive an operational response for the MOM. Additionally, the evaluation of operator performance is utilized, serving as the foundation for Key Performance Indicators (KPIs) and Human Resource Effectiveness (HRE), which are crucial components in the Knowledge Graph (KG) used to establish adaptable and resilient conditions for the operators. The expansion modules of the activity model are intricately linked to the KG through semantic technologies. The advancement of intelligent cyber-physical systems establishes an environment where each aspect of the intricate manufacturing system, involving humans and machines, is effectively supervised, and the information systems are compatible. The vital components of the extended MOM model, viewed from the shop floor perspective, include Operator Performance Analysis, HAR, and Monitoring, which plays a key role in KPI and metric evaluation. A thorough examination of operator performance can support skill-based matching and the development of skill clusters. With the growing need for adaptable production lines, conventional assembly lines might be substituted by self-sufficient work-stations, referred to as skill clusters, with mobile robots transitioning between them. In

addition, skill clusters should be furnished with collaborative robots capable of working safely and reliably alongside operators.

### 3.2. Human–Robot Collaboration and Key Performance Indicators

This section delves into the various categories of workstations and collaboration scenarios that are of significance within the context of the case study under consideration. In addition, a concise summary of the primary performance metrics related to human-centered, ergonomic, and human–robot collaboration is provided. Depending on whether the actors involved are human or robotic, different types of workstations can be identified: manual, collaborative, and automatic [53]. Within a collaborative workstation setup, further categorization is established based on the nature of the interaction between human and robot actors during work tasks, aiming to yield a more comprehensive case study. Figure 3 illustrates three different types of collaboration [42,54]:

1. Separate work: Human and robot tasks are kept apart, and they do not share workspaces, tools, or workpieces.
2. Sequential collaboration: Although the human and robot actors are in a shared process flow of a workpiece, tasks are completed in succession. The workspaces, tools, and workpieces may be shared, but the tasks are strictly serialized such that any sharing is temporally separated.
3. Simultaneous collaboration: Human and robot tasks are executed concurrently and, moreover, may involve working on different parts of the same workpiece but are focused on achieving separate task goals.
4. Supportive collaboration: Humans and robots work together on the same piece of work to complete a common task.

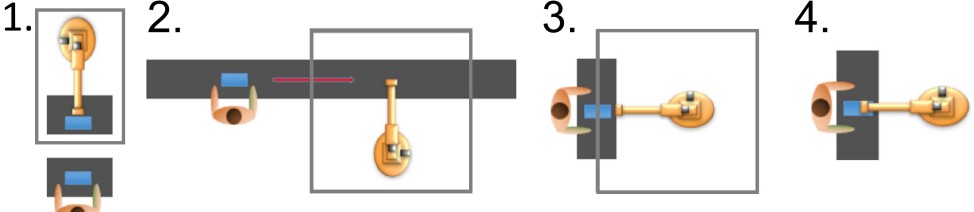

**Figure 3.** Separate (**1.**), sequential (**2.**), simultaneous (**3.**), and supportive (**4.**) types of human–robot collaborations [42].

The key performance indicators (KPIs) focused on human factors are outlined in Table 1 across six distinct categories: time behavior, physical measures, HR physical measures, efficiency, effectiveness, and ergonomics. Furthermore, the Operator 4.0 typologies [46] have been incorporated into these KPIs in the second column, illustrating the potential to facilitate the establishment of a harmonious relationship between humans and automation. A study that assesses the quality of human–robot interaction [15] has been partially referenced, although it does not offer a comprehensive overview of the subject matter. Additionally, a paper on ontology-driven KPI metamodeling [14] has been taken into account in this case study, focusing on a semantic technology perspective.

**Table 1.** The categorized human-centric KPIs for the case study.

| KPI Description | Operator 4.0 Type |
|---|---|
| Time behavior category | |
| Average time to complete task | Analytical operator |
| Collaboration time—Type-3 and Type-4 | Collaborative operator |
| Functional delays | Analytical operator |
| Human operation time | Analytical operator |
| Interaction time | Collaborative operator |

**Table 1.** *Cont.*

| KPI Description | Operator 4.0 Type |
|---|---|
| Response time | Collaborative operator |
| Robot functional delay | Collaborative operator |
| Robot operation time | Collaborative operator |
| Task completion time | Analytical operator |
| Total assembly time | Analytical operator |
| Total operation time | Analytical operator |
| Physiological measures category | |
| Biosignals (temperature, tactile, etc.) | Healthy operator |
| Ergonomics improvement | Healthy operator |
| Muscle activity | Healthy operator |
| Ocular behavior | Healthy operator |
| HR physical measures category | |
| Avg./min. length between a human hand and a robot hand | Collaborative operator |
| Human–robot distance | Collaborative operator |
| Efficiency category | |
| Availability | Collaborative operator |
| Average robot velocity | Collaborative operator |
| Concurrent activity | Collaborative operator |
| Degree of collaboration | Collaborative operator |
| Layout efficiency | Analytical operator |
| Effectiveness category | |
| Accuracy | Analytical operator |
| Interaction accuracy | Collaborative operator |
| Level of assignment | Collaborative operator |
| Level of interaction | Collaborative operator |
| Overall equipment effectiveness | Analytical operator |
| Real-time human fault | Analytical operator |
| Real-time robot fault | Collaborative operator |
| Ergonomics—environmental category | |
| Environmental condition—noise | Healthy operator |
| Environmental condition—humidity | Healthy operator |
| Environmental condition—temperature | Healthy operator |
| Environmental condition—gases | Healthy operator |

### 3.3. Design Structure of the HCKG Concept

This section outlines the methodology and presents a summary of the proposed development framework in a block format, as illustrated in Figure 4. The main objective is to integrate the human-centric KG block within a sophisticated industrial setting. The framework comprises five distinct blocks (or sections), commencing with the metadata sources from a business or industrial network and culminating in the application that leverages the information to generate value.

The meta block includes essential data to characterize the business processes and describable elements of a plant, such as material or information flows, starting from the foundational level. Markup languages and standards like B2MML (Business To Manufacturing Markup Language), AutomationML, or ISA-95 establish the initial framework for handling and overseeing the diverse data sources and processes within a complex network. It is advisable to expand existing standards such as ISA-95. A critical aspect of industrial progress involves utilizing standardized models, which facilitate the seamless integration of a new design concept into a production system and enhance the adaptability of existing methodologies, thereby making the learning curve for technical aspects more dynamic. The next component is the Schema and PPR block, representing the three descriptive on-

tologies within an Industry 4.0 setting. The product, process, and resource ontologies can comprehensively depict the entire network in a semantic format. Various assets, whether physical or human attributes, characteristics, and specific values, are structured as ontology axioms (individuals) and classified into classes. Furthermore, semantic properties, rules, and queries support interoperability and the depiction of relationships, such as actors' capabilities, the sequence of manufacturing activities, or resource allocation. PPR-based modeling aligns with AutomationML and serves as a method for establishing knowledge-driven mappings of products, processes, and resources in assembly automation [55]. The primary advantage of PPR-based modeling lies in facilitating the management of engineering datasets' mappings and connecting product attributes to manufacturing processes and resources. Moreover, knowledge-driven PPR mapping can aid in dynamically configuring and analyzing assembly automation systems [56]. The IoT Block comprises monitoring devices and sensors for conducting observations, along with HAR, which are essential inputs for the higher-level human-centric block—a pivotal component. Additionally, IoT devices form a complex system that necessitates separate management due to the diversity of smart devices and sensors. The so-called VAR ontology encompasses three key elements: tangible assets, intangible assets, and dynamic status.

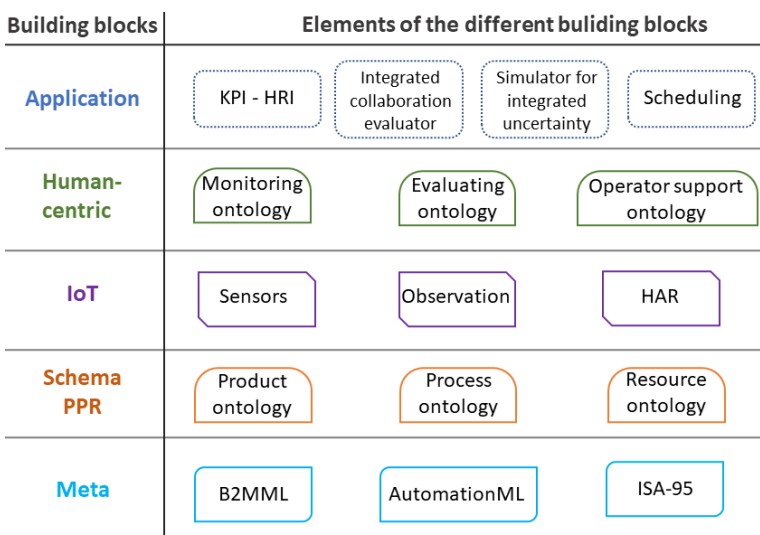

**Figure 4.** Theoretical structure of the proposed human-centered knowledge graph-based design concept [30].

The human-centric block includes the monitoring, evaluation, and operator support ontologies, with the goal of gathering and analyzing relevant information related to the production process, collaboration, human activities, or working conditions in the factory. Its primary aim is to keep the operator informed and assist them in various aspects such as ergonomics and collaboration. In a setting where humans and robots work together, feedback is crucial not only from the control or machine perspective but also from the human perspective, focusing on real ergonomic features, process parameters, and other input from the operator. Operators on the factory floor can offer valuable insights for the MOM's Operations Response, which should be integrated into the semantic-based data management system supporting CI/CD (Continuous Integration and Continuous Delivery) practices. The application block encompasses all the valuable information that the HCKG can provide for tasks like scheduling, resource allocation, enhancing KPIs and HRI factors, assessing collaboration aspects, or conducting simulations. The end user, whether a process engineer, factory worker, or production manager, is primarily interested in this segment as it delivers the final outcome of the semantic-based analysis. The application block can aid in exploring integrated uncertainty through simulations and assessing collaboration or business processes. Furthermore, scheduling and allocations can be optimized based on the performance metrics obtained. Other issues like the cybersecurity of large infrastructures,

which are not covered here, are likely to remain significant challenges in the foreseeable future. Subsequently, a case study demonstrating the application of the proposed KG framework is presented following the discussion on the KG design concept.

## 4. Human-Centered Knowledge Graph Representation for a Wire Harness Assembly Process

This section illustrates the implementation of the HCKG approach elaborated in Section 3. Initially, a case study specific to the industry is outlined in Section 4.1. The establishment and organization of the generated KG are thoroughly examined in Section 4.2. Lastly, Section 4.3 demonstrates the visualization and examination of the production data.

### 4.1. Wire Harness Assembly-Based Case Study

A recent study on research and development [57] emphasized the importance of exploring collaborative robots in wire harness assembly. The authors of this study also delved into the analysis and design of Intelligent Collaborative Manufacturing Spaces (ICMS) using a hypergraph-based approach similar to a referenced benchmark [58]. The wire harness assembly sector served as the inspiration for the case study discussed in this paper. Specifically, the case study focused on the manufacturing processes of a multinational wire harness assembly plant. Detailed information cannot be disclosed due to confidentiality policies; however, the proposed methodology is continually undergoing validation with manufacturing experts. Figure 5 illustrates the factory floor layout, featuring a coordinate system that establishes a grid for assigning operators and production resources like robots and machinery. The case study incorporates a real-time location system (RTLS) that monitors the whereabouts of assembly workers and assets. The X and Y axes on the shop floor correspond to potential RTLS-based positions. Distances required for material handling and transportation can be determined from the grid. Furthermore, the shop floor is divided into 18 distinct areas, for instance, ST_11, which can accommodate workstations.

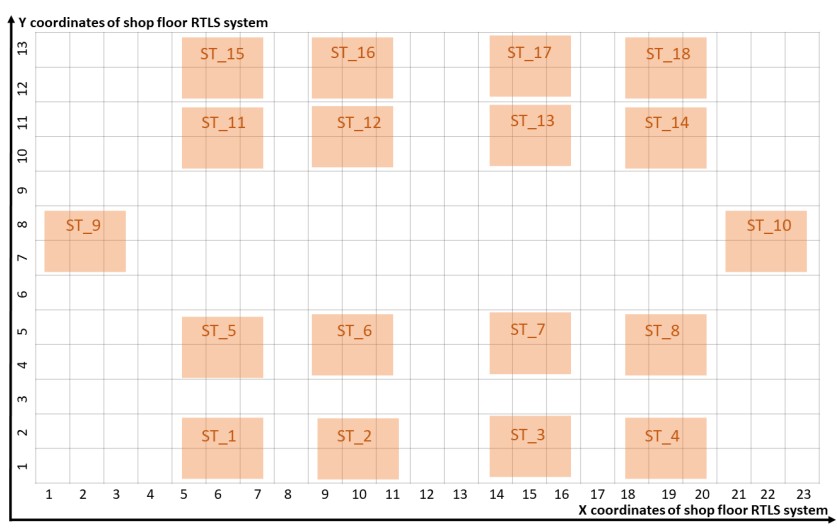

**Figure 5.** The grid layout of the benchmark shop floor.

A dual production system comprising batch and traditional production was specified, and the workflow is depicted in Figure 6, which is derived from an actual wire harness industry assembly line. This process involves two assembly lines that share tasks and resources. The components of these lines are detailed in Table 2. The shop floor includes two storage areas, multiple buffers, crimping stations, and assembly stations. The second group of components encompasses human–machine agents, which can be operators or robots, as well as production line assets, such as machines, tools, screwdrivers, and the AGV (Automated Guided Vehicle). Additionally, specific capabilities are necessary to perform designated tasks, along with sensor components, to oversee the collaborative environment.

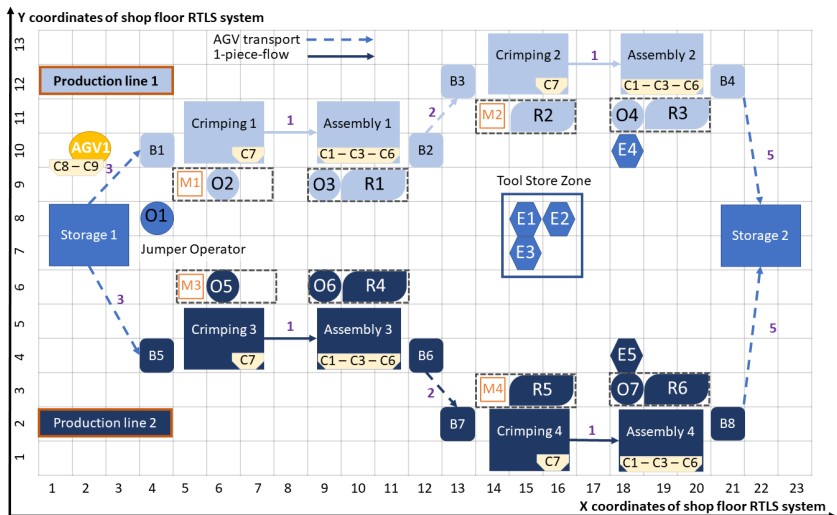

**Figure 6.** The process flow of the wire harness assembly line benchmark.

**Table 2.** The elements of the wire harness assembly lines.

| Work sections of the production lines | |
|---|---|
| Storage | [K1, K2] |
| Buffer | [B1, B2, B3, B4, B5, B6, B7, B8] |
| Crimping stations | [Crimping 1, Crimping 2, Crimping 3, Crimping 4] |
| Assembly stations | [Assembly 1, Assembly 2, Assembly 3, Assembly 4] |
| Human–machine members and assets | |
| Operators | [O1, O2, O3, O4, O5, O6, O7] |
| Robots | [R1, R2, R3, R4, R5, R6] |
| AGV | [AGV1] |
| Machines | [M1, M2, M3, M4] |
| Tools | [E1, E2, E3, E4, E5] |
| Capabilities | $[C1, C3, C6, C7, C8, C9]$ |

A detailed list of activity types for this benchmark problem can be found in Appendix A in Table A2, encompassing categories such as the crimping process, assembly process, or material handling, along with the definitions of the outcomes associated with these activity types. It is crucial not only to define the activity types but also their results for effective process tracking and collaboration. A more comprehensive overview of the wire harness assembly benchmark is presented in Appendix B, where each activity type of the intricate industrial process is outlined in Table A1, followed by detailed descriptions of the sequence of activities in Tables A3 and A4 of the Appendix B, as highlighted in this study. In addition to the main elements listed in Table 2, other attributes of the elements include the following Capabilities necessary for carrying out specific activities: $C1$—insertion and laying of parts (cabling), $C3$—terminal handling, $C6$—terminal screwing, $C7$—crimp machine operation, $C8$—loading or unloading of the $AGV$, and $C9$—workpiece transport on the shop floor. Moreover, there are specialized tools, some of which are shared across the process, namely $E1$—wiring tool, $E2$—hose tool, and $E3$–$E5$—screwdrivers. Additionally, various unique Machines (M) are allocated to different Crimping Stations, while Tools (E) are considered communal assets within Assembly Stations.

In Figure 6, the brighter-colored elements represent Production line 1, while the darker ones represent Production line 2. Shared assets and resources are visualized in the middle. Material handling steps during production are indicated with arrows, which can be carried out as a one-piece-flow by operators or through an AGV-based transport

system. Additionally, the distances over which materials are moved are marked with purple numbers. The process flow, illustrated in Figure 6, begins at Storage 1, where the operator known as jumper $O1$ loads $AGV1$ (using capability $C8$) with a batch, which is then transferred by $AGV1$ to either Crimping station 1 or 3 (using capability $C9$), where the unloading is performed by operator $O2$ or $O5$ into the local buffers $B1$ or $B5$. The subsequent steps are identical on both production lines, with the continuation of the process description focusing on Production line 1. As per the production plan, operator $O2$ carries out crimping-related activities listed in Table A2 that necessitate capability $C7$. Furthermore, machine $M1$ is utilized during these crimping activities. Subsequently, operator $O2$ transfers the workpiece to operator $O3$ at Assembly station 1 (one-piece-flow). Operator $O3$ and robot $R1$ collaborate, carrying out activities related to capabilities $C1$, $C3$, and $C6$. Additionally, tools $E1–3$ are employed during the activity steps at Assembly station 1. At the conclusion of the process, operator $O3$ places the workpiece into buffer $B2$. Upon the completion of a full batch, the same operator loads $AGV1$, which transports the batch of cables to the subsequent buffer, $B3$. Subsequently, robot $R2$ unloads the buffer and performs activities related to capability $C7$ and machine $M2$ at Crimping station 2. Following this, robot $R2$ transfers the workpiece (one-piece-flow) to operator $O4$ at the subsequent station, Assembly station 2. At the final workstation of Production line 1, operator $O4$ and robot $R3$ collaborate to carry out activities requiring capabilities $C1$, $C3$, and $C6$. At the end of the assembly line, operator $O4$ places the workpieces into buffer $B4$. Upon completion of a full batch, the same operator loads $AGV1$, which delivers the products to their final destination, Storage 2.

Furthermore, it is important to mention that this case study also includes different types of sensors, the purpose of which is to make observations about each activity, human, and machine of the production line, as well as to monitor the working conditions. These groups of sensors are camera systems, RTLS, robot-embedded sensor data, machine-embedded sensor data, environmental sensors, and human body sensors.

All three categories of workstations are included in this instance focusing on collaborative work. Crimping Station 1 and 3 are characterized as manual workstations, while Crimping Station 2 and 4 are classified as automatic workstations. The research features four collaborative workstations, denoted as Assembly stations 1–4. The specific case study on harness assembly delves into collaboration types 3 and 4. An illustration of concurrent and supportive collaboration is shown in Figure 7. In this simplified scenario, four distinct outcomes are observed, corresponding to the tasks executed by the entities Robot 3 and Operator 4. For instances Result 28 and 31, where both human and robot actors engage in similar activities on the same product, they engage in supportive collaborations aimed at achieving the same assembly result. On the contrary, Result 29 and 30 involve different activities, with human and robot actors operating on the same product simultaneously but pursuing different objectives. The idle period occurs when Robot 3 must wait for Operator 4.0 to complete their task, as they share the same workstation.

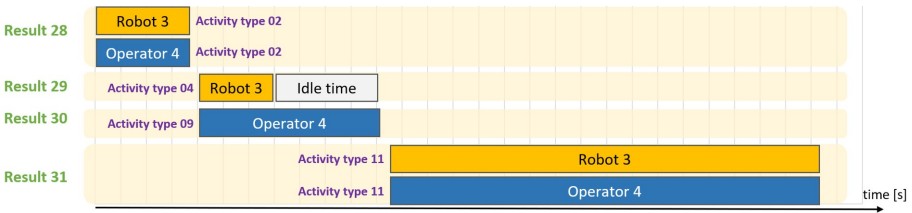

**Figure 7.** Gantt chart of collaboration scenarios.

## 4.2. Development of the Industry-Specific Human-Centered Knowledge Graph

A section of the KG that has been utilized in the case study discussed in Section 4.1 is depicted in Figure 8, excluding the distinct data properties of the ontology classes. The structural illustration in Figure 8 is segmented into six sets of ontology classes since the KG comprises multiple sub-ontologies. Moreover, the object properties, representing relationships among classes, are indicated on the arrows.

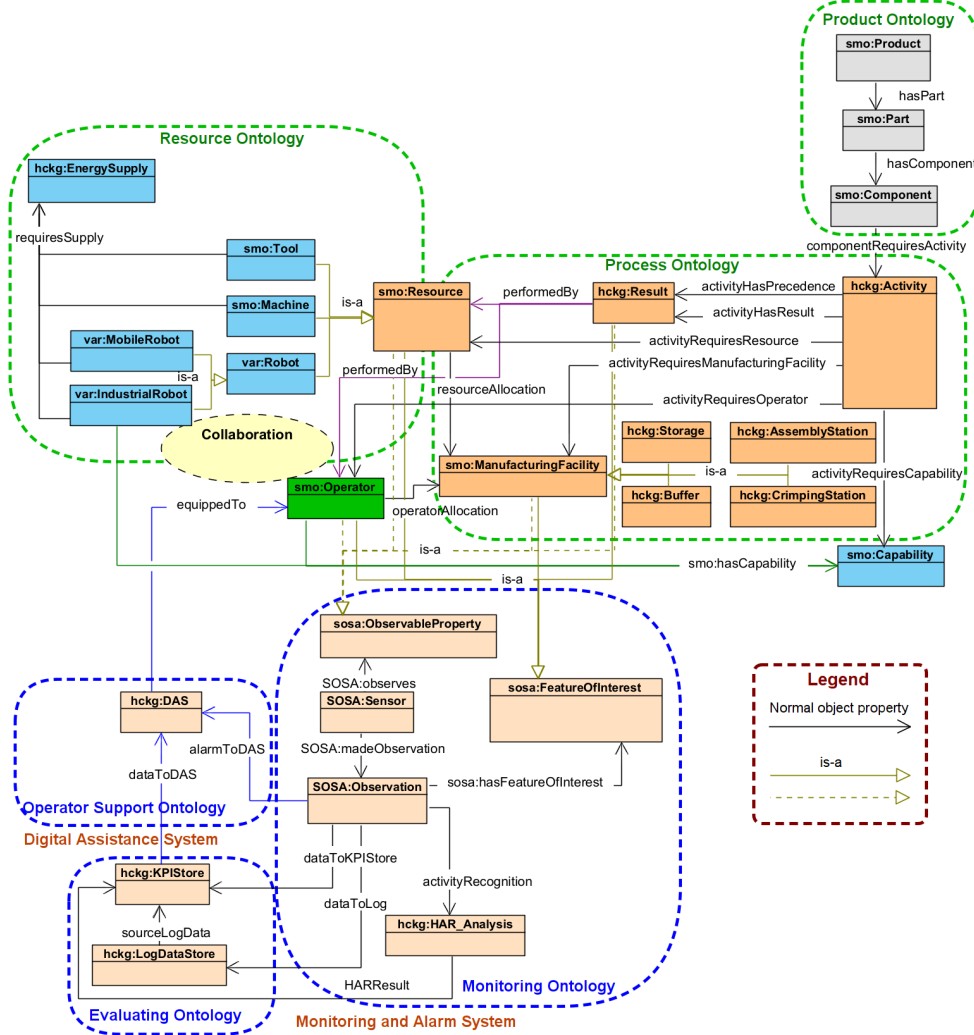

**Figure 8.** Partial structural diagram of the developed wire harness assembly-specific KG.

Prefixes indicating adapted namespaces from other industry-specific ontologies are included in the names of the ontology classes. The following list summarises these prefixes and the ontologies used:

- smo—Smart Manufacturing Ontology [59]: An ontology to model I4.0 production lines and smart factories based on RAMI 4.0. It highlights the sequence of processes and machines required for a produced workpiece.
- SOSA—Sensor, Observation, Sample, and Actuator ontology [60]: For modeling the interactions between the entities involved in terms of observation, actuation, and sampling. Together with SSN (Semantic Sensor Network), it can be used to describe sensors and their observations, the involved procedures, the studied features of interest, the samples used to do so, the feature's properties being observed or sampled, as well as actuators and the activities they trigger [61].
- var ontolog y [35]: A core ontology for data exchange in a semantic-oriented framework to support adaptive, interactive, assistive, and collaborative assembly workplaces.
- hckg—Human-Centric Knowledge Graph: The authors created a set of classes and properties to model the wire harness assembly-based case study semantically.

The product ontology includes three categories: product, part, and component. The complexity of this domain, previously addressed by the authors in [62], is not further explored here. The Process ontology is composed of the subsequent categories: Activity, Result, and ManufacturingFacility, which encompasses additional subclasses like Storage, Buffer, AssemblyStation, CrimpingStation, and Capability. The primary category in the

Resource ontology is Resource, which includes various subclasses such as Tool, Machine, and Robot. The Robot category is further divided into MobileRobot and IndustrialRobot. The EnergySupply category is also part of the resource ontology. The Operator category, a central element of the human-centered HC, is highlighted in green at the center of the HC structure in Figure 8. The operator category, which semantically characterizes the processes and impacts related to personnel on the shop floor, is associated with six distinct object properties. The Monitoring ontology comprises three categories: Sensor, Observation, and HAR_Analysis, storing the semantic model of sensor devices, their measurements, observation, and human activity recognition. The Evaluating ontology contains two categories: KPIStore and LogDataStore, designed to handle data from the aforementioned three categories. Lastly, in the Operator Support Ontology, the DAS category defines the digital assistance system. Considering that the categories Operator and Activity are pivotal in the KG, Tables 3 and 4 provide details on the corresponding object properties.

**Table 3.** Object properties of the Activity class.

| hckg:Activity | |
|---|---|
| component Requires Activity | Connects individuals from the component and activity classes and provides information about the required activity to assemble a specific component on the wire harness. |
| activity Has Precedence | Since the assembly procedure requires a specific sequence, certain activities must be finished before another can be started. This is known as the precedence criteria. |
| activity Has Result | Describes the intended result of a particular activity. In the case of collaboration, several activity individuals may be connected to the same result individual. |
| activity Requires Resource | Interlinks Tool, Machine, or Robot individuals to an activity as a resource requirement. |
| activity Requires—Manufacturing Facility | Workstation requirement of an activity. Connects activity individuals with the ManufacturingFacility individuals such as Storage, Buffer, AssemblyStation, or CrimpingStation. |
| activity Requires Operator | Connects operator individuals to an activity as a personnel requirement. |
| activity Requires Capability | Describes the capability requirement of a specific assembly activity, which has to be conducted by an Operator or IndustrialRobot. |

**Table 4.** Object properties of the Operator class.

| smo:Operator | |
|---|---|
| activity Requires Operator | It provides information about a certain operator involved in certain activities. |
| operator Allocation | Semantically connects operators with Manufacturing Facility individuals such as Storage, Buffer, AssemblyStation, or CrimpingStation. It provides information on where the operator performs his/her work. |
| performed By | Connects Results with Operators and shows which operator was involved in which result(s). |
| equipped To | Describes the usage of devices from the Digital Assistance System by operators. |

**Table 4.** *Cont.*

| smo:Operator | |
|---|---|
| is-a SOSA: Observable Property | Semantically connects the properties, which are monitored by sensors with operators and shows how personnel are monitored. |
| is-a SOSA: Feature Of Interest | Main class of the feature of interest of the SOSA:Observation |
| smo:hasCapability | Shows which capabilities require a specific operator. |

In addition, some of the key features of the use of semantic technologies and graph analysis from a human-centered approach are presented in  Table 5 [63]. These analytics can help to better monitor and understand the HRE [64] and KPI [65] factors. In addition, Table  5 provides an example of its application for each network metric.

**Table 5.** KG metrics and analytical features.

| Network Metrics | Analytical Features of KGs |
|---|---|
| Centrality computation | Which are the critical objects in the network? |
| | Detect the most significant influencing factors in the operator's environment. |
| Similarities between nodes and edges | How similar are two objects based on their properties and how are they connected to other objects? |
| | Solve allocation problems concerning operators and resources. |
| Flows and paths | What is the shortest, cheapest, or quickest way to perform a process step? |
| | Optimise the shop floor layout to best match operator needs. |
| Cycles | Are there any cycles in the graph? If so, where are they? |
| | Analyze tasks allocated to humans and machines in a collaborative work environment |
| Network communities | What communities can be found in the production network? |
| | Facilitate the design of human–machine collaboration or cell formation. |

Once the use case-specific knowledge graph has been established and the necessary data have been imported into the semantic network, the subsequent stage involves formulating queries and examining the outcomes. Consequently, the subsequent subsection delves into the examples of graph-based knowledge analyses that were utilized.

### 4.3. Discussion on KG-Based Analytics of the Use Case

Initially, Figure 9 shows the graphical representation of the complete knowledge graph (KG) related to the wire harness assembly case study. This visual depiction serves as a means to validate the manufacturing process. The complete network is illustrated on the left side, encompassing all properties and entities within the KG, while a more detailed view is presented on the right side. The orange node corresponds to the equipment $E5$ and includes various associated data properties like locationID (18-4), equipmentCondition (86), equipmentID ($E5$), equipmentName (screwdriver C), and equipmentType (screwdriver).

All the SPARQL queries that have been developed are accessible on our website at https://github.com/abonyilab/HCKG (accessed on 14 March 2024). The initial instance of a SPARQL query-based data mapping, as referenced in [66], is illustrated in Figure 10. The left section of Figure 10 shows a graphical representation of the query, where four distinct rules are outlined to achieve the intended result. This particular example aims to identify RobotAssets categorized as IndustrialRobot s and seeks to present three associated data elements: Location , EnergySupply , and ManufacturingFacility. On the right side of Figure 10, a graphical representation of the result of the query is provided.  Robot

assets classified as IndustrialRobot s are depicted as orange nodes, each necessitating an EnergySupply referred to as g2 (electricity). Each IndustrialRobot node is linked to the corresponding workstation node (ManufacturingFacility ), which represents various assembly stations, illustrated in purple in this scenario. Lastly, the location data attributes of the robots are denoted by blue nodes (indicating the robots require two zones on the shop floor). This type of visual examination can aid in exploring dependencies concerning specific assets.

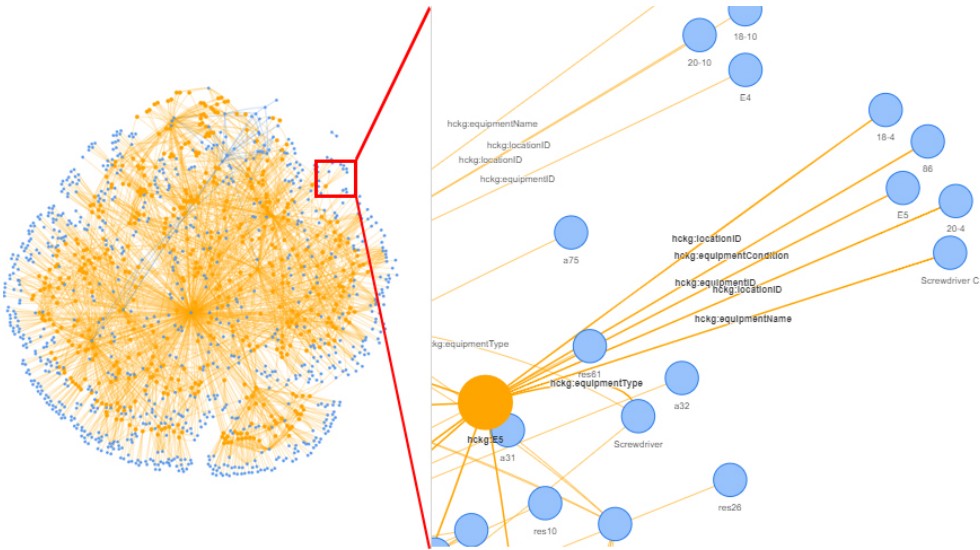

**Figure 9.** Visualization of the entire KG of this case study (on the left-hand side) and some of the data properties of equipment *E*5 (on the right-hand side—zoom into the red rectangle).

The resulting graph in Figure 11 illustrates the connections between Actors (operators or robots) and the Capability entities they are linked to. This illustration can be utilized for a visual examination of production capability. In Figure 11, it is evident that the majority of actors possess the capability *C*8 (AGV loading/unloading). Moreover, robots are limited to a maximum of two capabilities, whereas operators can concurrently possess up to four capabilities.

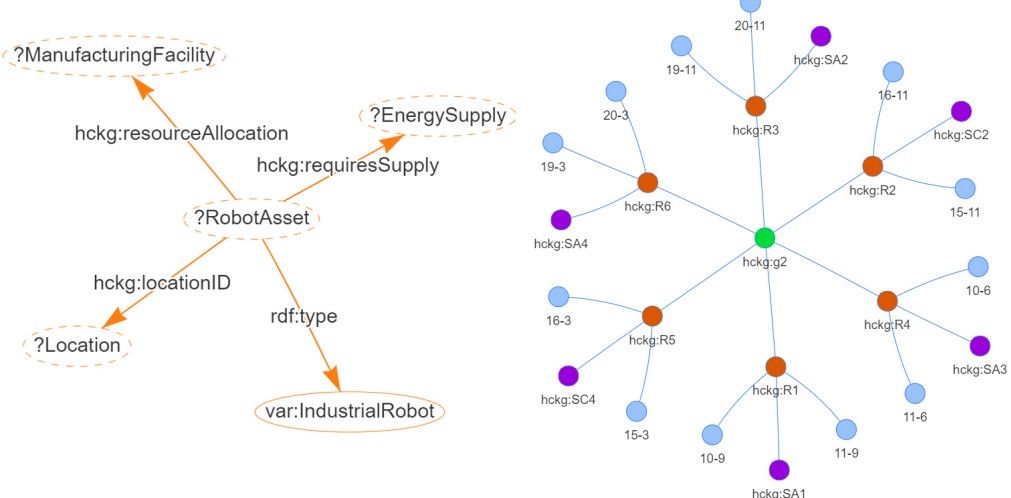

**Figure 10.** Visualization of the RobotAsset query (on the left-hand side) and the graph visualization of the result (on the right-hand side).

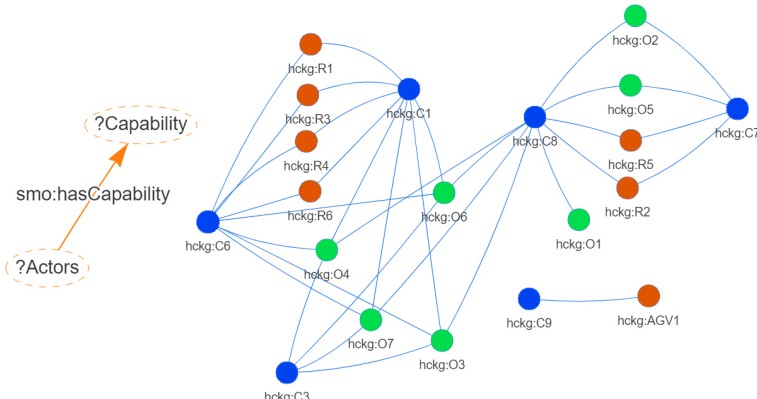

**Figure 11.** Visualization of the Actors-Capability query (on the left-hand side) and the graph visualization of the result (on the right-hand side).

A more intricate data query aimed at identifying sensor alerts transmitted to DAS devices is outlined in Figure 12. Initially, the KG is streamlined to include only the sensor, observation, and observed nodes, which are then refined to encompass sensor instances categorized under type names starting with "env" or "body", denoting environmental or body sensors. Subsequently, additional data are integrated into the dataset, specifying attributes such as observationValue, warningLimit, and alarmLimit. A subsequent filter is employed to isolate instances where the observationValue exceeds the warningLimit. The output comprises a compilation of the DAS device name, the message content, and the device's location. On the right-hand side of Figure 12, a graphical representation showcases only the most pertinent segment of the query outcome. Here, the purple nodes symbolize the sensor's location, the red nodes depict the message relayed to the DAS, and the green nodes correspond to the specific operator integrated into the DAS device, such as Smart Glass. Notably, in the graph located in the lower right corner of the figure, the locations of the observation sensor and the DAS device coincide, indicating that they are body sensors.

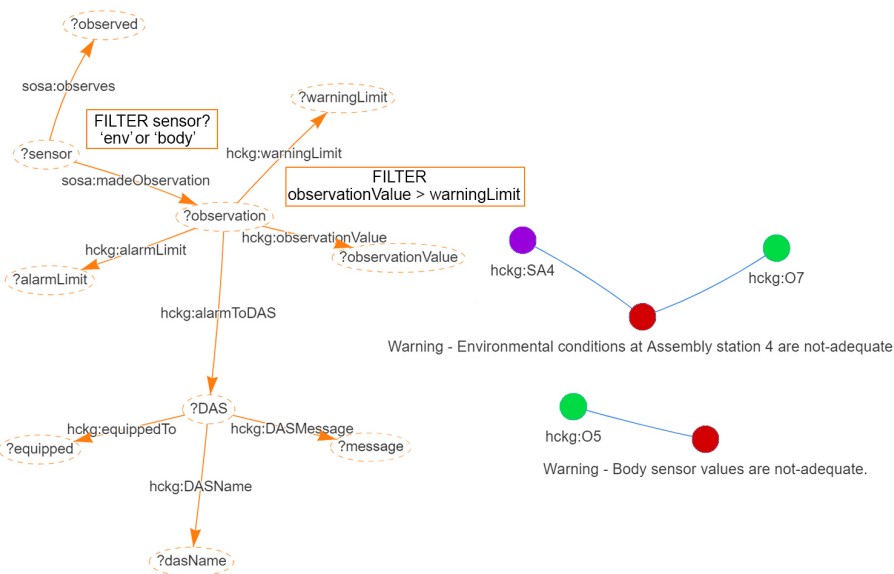

**Figure 12.** Visualization of the sensor-observation-DAS query (on the left-hand side) and a graph visualization of the result (on the right-hand side).

In the context of Operator 7 (*O7*) and Robots 6–7 (*R6 − R7*) collaborating at Assembly Station 4, as depicted in Figures 13 and 14 illustrates the application of the time KPI for human–machine collaboration. The graph in Figure 14 displays the total duration of supportive collaboration (type 4). Notably, *O7* dedicated more time to collaborative

assembly with Robots 5 and 6 than to individual tasks. Furthermore, the graph highlights simultaneous collaboration (type 3), particularly between Operator 7 and Robot 6 in the last two columns. To analyze both type 3 and the sequence of concurrent collaborative assembly actions, it is essential to examine the outcomes and precedence of these activities. Consequently, Figure 15 presents the results of a knowledge graph query visualized through directed graphs, depicting precedence relationships. In these graphs, yellow nodes signify the activities, while purple nodes represent the outcomes.

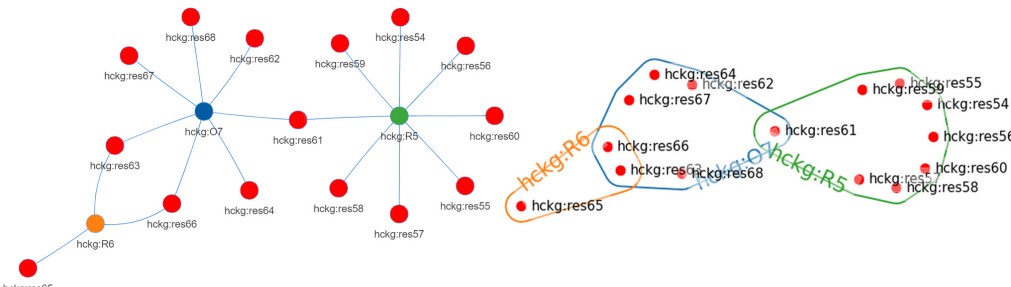

**Figure 13.** Visualization of human–robot actors and the performed results at Assembly station 4 in the form of a graph (on the left-hand side) and hypergraph (on the right-hand side).

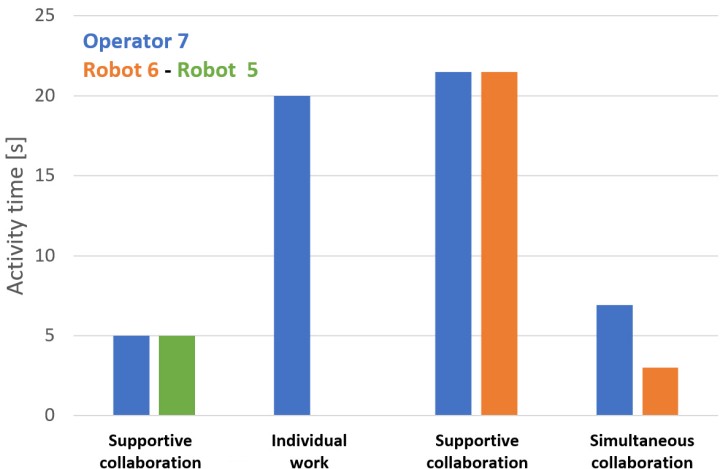

**Figure 14.** Distribution of assembly work in terms of operator *O7*, including the total supportive, simultaneous, and individual times.

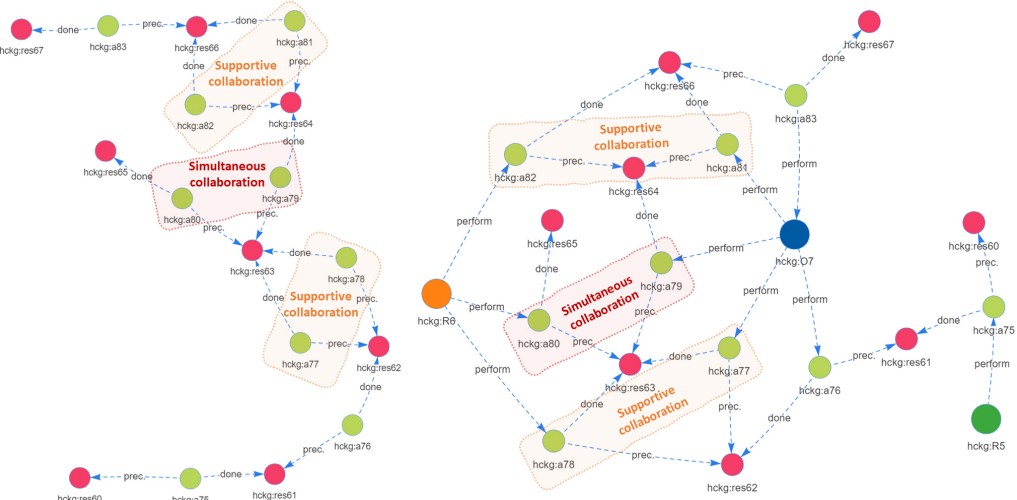

**Figure 15.** Directed graph result and activity nodes (on the left-hand side) as well as the same result, including the human–machine actor nodes (on the right-hand side).

The directed edges represent different object properties of the KG, namely:

- done—activityHasResult object property: shows the result condition of a specific activity if the assembly task is performed.
- prec.—activityHasPrecedence object property: represents the precedence criteria of an activity that has to be carried out before the specific activity can be started.
- perform—performedBy object property: describes the human or robot actor that performs the activity.

The tasks and outcomes that serve as a foundation for analyzing the process flow, where the order of steps and conditions can be traced from tasks $a75$ to $a83$, are presented on the left side of Figure 15. An expanded visualization, which includes the perform connections indicating that a human or robotic agent has performed a specific task, is shown on the right side of the same figure. Examining the inbound and outbound connections of a directed graph [67] enables the identification of clusters [68] within the network. By applying this approach, it can be inferred that if a result node has multiple completed incoming connections, it has been carried out through a collaborative effort involving actors of type-4 support, as indicated in the instances of tasks $a77 - a78$ and $a81 - a82$. In such scenarios, the actors are required to wait for the completion of the same outcome (priority is given) before commencing different tasks simultaneously on the same work item.

According to the precedence graph, when two or more activity nodes are assigned the same precedence (prec. edge) but lead to different outcomes (done edge), it indicates a type-3 concurrent collaboration. In Figure 15, it is evident that activities $a79$ and $a80$ are executed simultaneously after receiving the same precedence (res63), yet they generate distinct results upon completion (res64 and res65). The outcome of this section is a conceptual dashboard, shown in Figure 16, where the percentages indicate the levels of operator competence and robot health. The findings from the previous query, along with the KPIs in Section 3.2, can serve as data sources for smart glasses, shop floor dashboards, the DAS, or other intelligent devices.

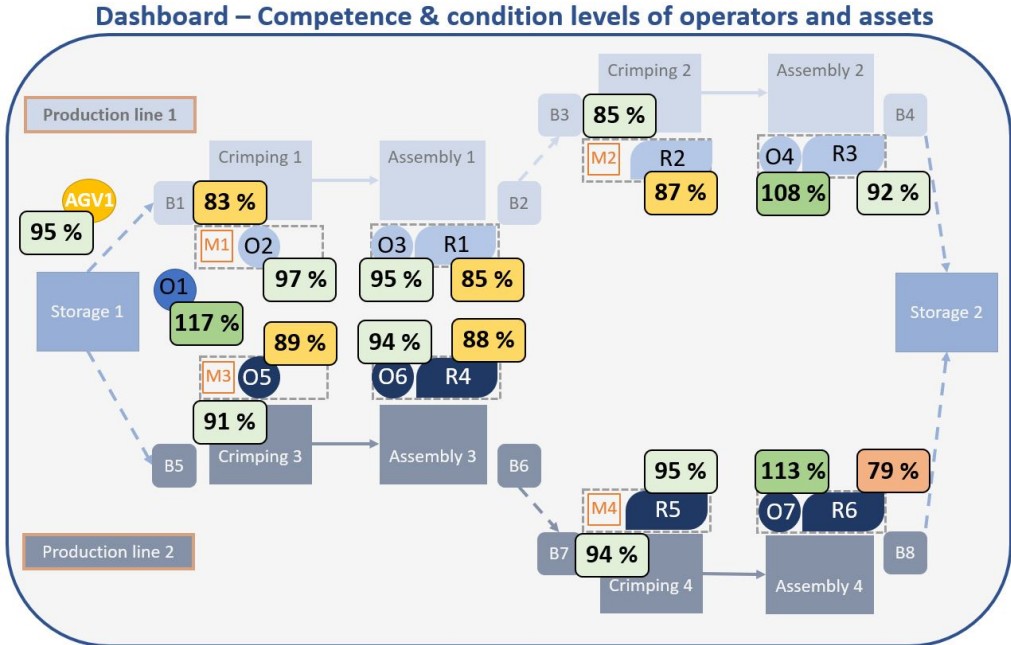

**Figure 16.** Conceptual dashboard for human-centric manufacturing—competence and condition levels of operators and assets.

## 5. Conclusions and Future Work

This article introduces the design idea of a Human-Centred Knowledge Graph (HCKG) that is based on industry norms and semantic technologies related to Industry 5.0 advance-

ments. A block structure is presented as an enhanced version of the MOM model and the development framework. The study thoroughly considers the tasks performed by operators, encompassing movement assessment, collaboration with machines, work sequences, and ergonomic aspects. It is also emphasized that the integration of activity recognition technologies can enrich the valuable data within a knowledge graph in a smart factory setting. The issue of insufficient operator monitoring and assistance is discussed in the context of existing industry standards, advocating for a new human-centric approach to contemporary manufacturing practices. In the coming factories that use knowledge graphs, the data collection and knowledge exploration processes will be automated, thereby facilitating the creation of human digital twins and the adoption of Industry 5.0 technologies.

Our objective was to summarize current methods and tools for semantic development and to introduce a concept for creating standard models of human-centered collaboration, illustrated through an industrial case study. The key contributions of this paper are as follows.

- Emphasized the importance of incorporating human factors into cyber-physical systems.
- Proposed an expansion of automation standards (ISA-95, AutomationML, B2MML) to include human-related processes and demonstrated the use of semantic technologies.
- The concept was validated through a replicable industrial case study. Various graph-based analyses were conducted using different types of graphs such as normal, directed, or hypergraphs, including resource allocation analysis, KPI evaluation, and the integration of a DAS.
- The application based on HCKG facilitated the identification of various forms of collaboration between human and machine actors in the assembly process.
- Furthermore, a conceptual design was put forward for a human-centric manufacturing dashboard.

Future research will focus on enhancing the design of human–machine and human–human collaboration in manufacturing by implementing the HCKG concept in an intelligent environment. Several areas for improvement should be considered in future studies. One aspect is the incorporation of additional functionalities within the application block, such as an uncertainty simulator, a collaboration assessment tool, or an intelligent scheduling mechanism. Moreover, integrating HCKG into a digital twin and implementing closed-loop optimization and decision support could further strengthen the proposed approach. Lastly, it is crucial to encompass the entire HCKG pipeline and establish automated data retrieval within the shop floor and the semantic network.

**Author Contributions:** Conceptualization, J.A. and L.N.; methodology, L.N. and T.R.; validation, T.R. writing—original draft preparation, L.N. and T.R.; writing—review and editing, J.A.; visualization, L.N.; supervision, J.A. All authors have read and agreed to the published version of the manuscript.

**Funding:** This work has been implemented by OTKA 143482 (Monitoring Complex Systems by goal-oriented clustering algorithms) with support provided by the National Ministry of Hungary for Culture and Innovation from the National Research, Development, and Innovation Fund. L.N. was supported by the ÚNKP-22-3 New National Excellence Program of the Ministry for Culture and Innovation from the source of the National Research, Development, and Innovation Fund.

**Institutional Review Board Statement:** Not applicable.

**Informed Consent Statement:** Not applicable.

**Data Availability Statement:** The related dataset is freely and fully available on the website of the authors: https://github.com/abonyilab/HCKG (accessed on 14 March 2024).

**Conflicts of Interest:** The authors declare no conflicts of interest.

## Appendix A. Applied Methodologies and Software Tools

Figure A1 shows several processing stages of a data pipeline based on a study [69] that aims to create KGs for the automation industry. In addition, an end-to-end digital twin pipeline [70] has been taken into account.

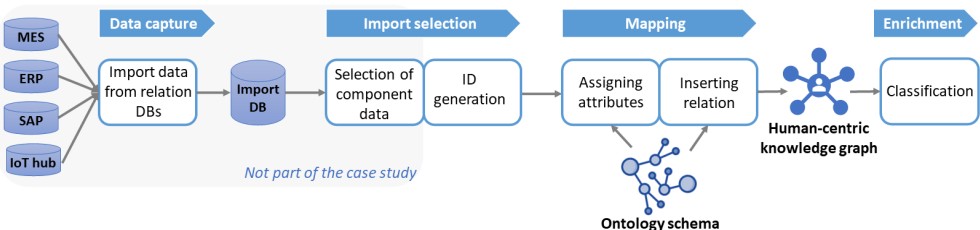

**Figure A1.** Knowledge graph pipeline based on [69].

It is beyond the scope of this paper to discuss the data acquisition and import selection parts of the pipeline. Only the KG, the ontology creation, the data queries, the mapping, and the data enrichment and visualization will be discussed. The phases, the methods used, and the different software stages of the industrial case study presented are shown in Figure A2.

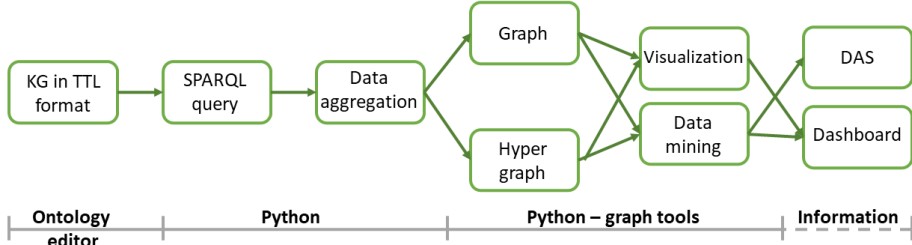

**Figure A2.** The steps of the applied method.

Before processing the TTL file in a Python 3.12.3 environment using Pyvis (a Python library for visualizing networks) [71] and KGlab [72,73], the sub-ontologies and the whole KG were developed using Protégé [74]. Either Protégé or KGlab can be used to import data into the ontology skeleton and to create axioms and properties in Python. For each data query, SPARQL language has been used [66]; moreover, Pyvis offers a graphical representation. Having mapped the semantic data, it was further aggregated in Python to obtain data-enriched graphs for analysis. Normal, directed, or hypergraphs can also be used for graph-based visualization of KG data. Finally, as a concept (denoted by a dashed line in Figure A2), the key information, generated charts, statements, or messages can be displayed on dashboards and DAS devices, or fill any other elements of the application block with data, as previously presented in Figure 4.

## Appendix B. Assembly Activities of the Wire Harness Production Benchmark

**Table A1.** Description of the different activity types throughout the wire harness assembly benchmark.

| Activity Type ID | Description of the Activity Type |
| :---: | :--- |
| t1 | Point-to-point wiring on a chassis |
| t2 | Laying in a U-channel |
| t3 | Laying a flat cable |
| t4 | Laying wire(s) onto the harness jig |
| t5 | Laying one end of a cable connector onto a harness jig |
| t6 | Spot-tying onto a cable and cutting it with a pair of scissors |
| t7 | Lacing activity |
| t8 | Lacing activity |
| t9 | Inserting into a tube or sleeve |
| t10 | Attachment of a wire terminal |
| t11 | Screw fastening of a wire terminal |
| t12 | Screw-and-nut fastening of a wire terminal |
| t13 | Circular connector |
| t14 | Rectangular connector |
| t15 | Clip installation |

**Table A1.** *Cont.*

| | Activity Type ID | Description of the Activity Type |
|---|---|---|
| | t16 | Loading of the AGV |
| | t17 | Transportation |
| | t18 | Manual handling of a wire from a buffer |
| | t19 | Positioning of a crimp into a vise |
| | t20 | Inserting a wire into a crimp |
| | t21 | Starting a machine |
| | t22 | Crimping |
| | t23 | Manual handling of a semi-finished product |
| | t24 | Handover of a semi-finished product |
| | t25 | Positioning of a crimp into a fixture |
| | t26 | Manual handling of a semi-finished product into a buffer |
| | t27 | Unloading of the AGV |

**Table A2.** The activity types in the wire harness assembly process and their results.

| | Crimping Process |
|---|---|
| t18 | Manual handling of a wire from a buffer<br>Result: One piece of wire is moved to the crimping station from the buffer. |
| t19 | Positioning of a crimp into a vise<br>Result: Crimp is positioned into a vise. |
| t20 | Inserting a wire into a crimp<br>Result: Wire is inserted into a crimp. |
| t21 | Starting a machine<br>Result: Machine is running. |
| t22 | Crimping<br>Result: Crimping is finished. |
| t23 | Manual handling of a semi-finished product<br>Result: Semi-finished product is removed from the vise. |
| t24 | Handover of a semi-finished product<br>Result: Semi-finished product is moved to another station. |
| | **Assembly process** |
| t2 | Laying in a U-channel<br>Result: U-channel is laid in the right assembly zone. |
| t4 | Laying wire(s) onto a harness jig<br>Result: Wire(s) is (are) laid correctly onto a harness jig. |
| t9 | Insertion into a tube or sleeve<br>Result: Tube is inserted into the correct sleeve. |
| t11 | Fastening of the terminal with screws<br>Result: Terminal screws are fastened. |
| t25 | Positioning of a crimp into a fixture<br>Result: Crimp is correctly positioned into the fixture. |
| t26 | Manual handling of a semi-finished product into a buffer<br>Result: Semi-finished product is placed into the buffer. |
| | **Material handling** |
| t16 | Loading of the AGV<br>Result: Parts are loaded on to the rack of the AGV. |
| t17 | Transportation by an AGV<br>Result: AGV moved the position from the source to its destination |
| t27 | Unloading of the AGV<br>Result: Parts are unloaded from the rack of the AGV. |

**Table A3.** The sequence of activities as well as the results of the proposed wire harness assembly benchmark and their details—Part 1.

| Activity ID | Activity Type ID | Result ID | Result Type ID | Process Step | Number of Process Steps |
|---|---|---|---|---|---|
| a1 | t16 | res1 | res_type_16 | Storage 1—AGV1 | 1 |
| a2 | t17 | res2 | res_type_17 | Storage 1—Buffer1 | 1 |
| a3 | t27 | res3 | res_type_27 | AGV1—Buffer1 | 1 |
| a4 | t18 | res4 | res_type_18 | Buffer1—Crimping1 | Batch size |
| a5 | t19 | res5 | res_type_19 | Crimping1 | Batch size |
| a6 | t20 | res6 | res_type_20 | Crimping1 | Batch size |
| a7 | t21 | res7 | res_type_21 | Crimping1 | Batch size |
| a8 | t22 | res8 | res_type_22 | Crimping1 | Batch size |
| a9 | t23 | res9 | res_type_23 | Crimping1 | Batch size |
| a10 | t24 | res10 | res_type_24 | Crimping1—Assembly1 | Batch size |
| a11 | t24 | res10 | res_type_24 | Crimping1—Assembly1 | Batch size |
| a12 | t25 | res11 | res_type_25 | Assembly1 | Batch size |
| a13 | t02 | res12 | res_type_02 | Assembly1 | Batch size |
| a14 | t02 | res12 | res_type_02 | Assembly1 | Batch size |
| a15 | t04 | res13 | res_type_04 | Assembly1 | Batch size |
| a16 | t04 | res13 | res_type_04 | Assembly1 | Batch size |
| a17 | t09 | res14 | res_type_09 | Assembly1 | Batch size |
| a18 | t09 | res14 | res_type_09 | Assembly1 | Batch size |
| a19 | t11 | res15 | res_type_11 | Assembly1 | Batch size |
| a20 | t11 | res15 | res_type_11 | Assembly1 | Batch size |
| a21 | t26 | res16 | res_type_26 | Assembly1—Buffer2 | Batch size |
| a22 | t16 | res17 | res_type_16 | Buffer2—AGV1 | 1 |
| a23 | t17 | res18 | res_type_17 | Buffer2—Buffer3 | 1 |
| a24 | t27 | res19 | res_type_27 | AGV1—Buffer3 | 1 |
| a25 | t18 | res20 | res_type_18 | Buffer3—Crimping2 | Batch size |
| a26 | t19 | res21 | res_type_19 | Crimping2 | Batch size |
| a27 | t20 | res22 | res_type_20 | Crimping2 | Batch size |
| a28 | t21 | res23 | res_type_21 | Crimping2 | Batch size |
| a29 | t22 | res24 | res_type_22 | Crimping2 | Batch size |
| a30 | t23 | res25 | res_type_23 | Crimping2 | Batch size |
| a31 | t24 | res26 | res_type_24 | Crimping2—Assembly2 | Batch size |
| a32 | t24 | res26 | res_type_24 | Crimping2—Assembly2 | Batch size |
| a33 | t25 | res27 | res_type_25 | Assembly2 | Batch size |
| a34 | t02 | res28 | res_type_02 | Assembly2 | Batch size |
| a35 | t02 | res28 | res_type_02 | Assembly2 | Batch size |
| a36 | t04 | res29 | res_type_04 | Assembly2 | Batch size |
| a37 | t09 | res30 | res_type_09 | Assembly2 | Batch size |
| a38 | t11 | res31 | res_type_11 | Assembly2 | Batch size |
| a39 | t11 | res31 | res_type_11 | Assembly2 | Batch size |
| a40 | t26 | res32 | res_type_26 | Assembly2—Buffer4 | Batch size |
| a41 | t16 | res33 | res_type_16 | Buffer4—AGV1 | 1 |
| a42 | t17 | res34 | res_type_17 | Buffer4—Buffer9 | 1 |
| a43 | t27 | res35 | res_type_27 | AGV1—Storage 2 | 1 |
| a44 | t16 | res36 | res_type_16 | Storage 1—AGV1 | 1 |
| a45 | t17 | res37 | res_type_17 | Storage 1—Buffer5 | 1 |
| a46 | t27 | res38 | res_type_27 | AGV1—Buffer5 | 1 |
| a47 | t18 | res39 | res_type_18 | Buffer5—Crimping3 | Batch size |
| a48 | t19 | res40 | res_type_19 | Crimping3 | Batch size |
| a49 | t20 | res41 | res_type_20 | Crimping3 | Batch size |
| a50 | t21 | res42 | res_type_21 | Crimping3 | Batch size |
| a51 | t22 | res43 | res_type_22 | Crimping3 | Batch size |
| a52 | t23 | res44 | res_type_23 | Crimping3 | Batch size |
| a53 | t24 | res45 | res_type_24 | Crimping3—Assembly3 | Batch size |
| a54 | t24 | res45 | res_type_24 | Crimping3—Assembly3 | Batch size |
| a55 | t25 | res46 | res_type_25 | Assembly3 | Batch size |
| a56 | t02 | res47 | res_type_02 | Assembly3 | Batch size |
| a57 | t02 | res47 | res_type_02 | Assembly3 | Batch size |
| a58 | t04 | res48 | res_type_04 | Assembly3 | Batch size |
| a59 | t04 | res48 | res_type_04 | Assembly3 | Batch size |
| a60 | t09 | res49 | res_type_09 | Assembly3 | Batch size |
| a61 | t09 | res49 | res_type_09 | Assembly3 | Batch size |
| a62 | t11 | res50 | res_type_11 | Assembly3 | Batch size |
| a63 | t11 | res50 | res_type_11 | Assembly3 | Batch size |
| a64 | t26 | res51 | res_type_26 | Assembly3—Buffer6 | Batch size |
| a65 | t16 | res52 | res_type_16 | Buffer6—AGV1 | 1 |
| a66 | t17 | res53 | res_type_17 | Buffer6—Buffer7 | 1 |
| a67 | t27 | res54 | res_type_27 | AGV1—Buffer7 | 1 |
| a68 | t18 | res55 | res_type_18 | Buffer7—Crimping4 | Batch size |
| a69 | t19 | res56 | res_type_19 | Crimping4 | Batch size |
| a70 | t20 | res57 | res_type_20 | Crimping4 | Batch size |
| a71 | t21 | res58 | res_type_21 | Crimping4 | Batch size |
| a72 | t22 | res59 | res_type_22 | Crimping4 | Batch size |
| a73 | t23 | res60 | res_type_23 | Crimping4 | Batch size |

**Table A4.** The sequence of activities as well as the results of the proposed wire harness assembly benchmark and their details—Part 2.

| Activity ID | Activity Type ID | Result ID | Result Type ID | Process Step | Number of Process Steps |
|---|---|---|---|---|---|
| a74 | t24 | res61 | res_type_24 | Crimping4—Assembly4 | Batch size |
| a75 | t24 | res61 | res_type_24 | Crimping4—Assembly4 | Batch size |
| a76 | t25 | res62 | res_type_25 | Assembly4 | Batch size |
| a77 | t02 | res63 | res_type_02 | Assembly4 | Batch size |
| a78 | t02 | res63 | res_type_02 | Assembly4 | Batch size |
| a79 | t04 | res64 | res_type_04 | Assembly4 | Batch size |
| a80 | t09 | res65 | res_type_09 | Assembly4 | Batch size |
| a81 | t11 | res66 | res_type_11 | Assembly4 | Batch size |
| a82 | t11 | res66 | res_type_11 | Assembly4 | Batch size |
| a83 | t26 | res67 | res_type_26 | Assembly4—Buffer8 | Batch size |
| a84 | t16 | res68 | res_type_16 | Buffer8—AGV1 | 1 |
| a85 | t17 | res69 | res_type_17 | Buffer8—Buffer9 | 1 |
| a86 | t27 | res70 | res_type_27 | AGV1—Storage 2 | 1 |

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
