# Peer review of "Knowledge Graph-Based Framework to Support Human-Centered Collaborative Manufacturing in Industry 5.0"

_applsci, doi:10.3390/app14083398_

Round 1

Reviewer 1 Report

Comments and Suggestions for Authors

Manuscript Number: applsci-2930152-peer-review-v1

Title: Knowledge graph-based framework to support human-centered collaborative manufacturing in Industry 5.0

The term Industry 4.0 refers to the integration of automation and data exchange in manufacturing. Industry 5.0 is a new concept that focuses on collaboration between humans and machines. The goal is to create sustainable products and services.

This study focuses on a Human-Centric Knowledge Graph framework by adapting Industry 5.0 standards to model the operator-related factors such as monitoring movements, working conditions, or collaborating with robots. The main contribution of this work is a knowledge graph-based framework that focuses on the work performed by the operator, including the evaluation of movements, collaboration with machines, ergonomics, and other conditions.

The novelty of this paper is highlighted well. It requires a revision to be accepted/rejected. Please see my comments below.

- Some concepts required rigid definitions. For example, “Meta” is appeared in words such as meta element, meta information, meta data, metamodeling, and meta block. I wonder what is the meaning of meta here?

- It is good to have some examples in such a framework to clarify some concepts and activities. Figure 7 is one of them and I believe it is a good Gantt chart example where human and robot spend same time on a product in Result 28 and 31. However, Result 29 and 30 are different. It would be nice if authors can further explain the meaning if the “idle time” for such a case.

- Authors should not use both British and American spelling in one article. Please do not mix the two in a single piece of writing. I have observed both of them in the paper like: utilise, utilize, behaviour, behavior, organise, organize, summarise, summarize, recognise, recognize, characterise, characterize, minimise, minimize, optimise, optimize.

- In written research papers, titles are not used on figures or tables; instead, the information is placed in a caption. Captions for tables are placed above the table (typically left aligned), and captions for figures are placed below the figure. Accordingly, authors should place captions for tables above the tables in this paper.

- Figure 3 shows three different types of collaboration including Sequential (2.), simultaneous (3.) and supportive (4.) types of human-robot collaborations. However, I believe that adding (1.) Separate work to this figure can make it more comprehensive.   

- Figure 16 is in the conclusion section. Please move it to the top of this section to have a conclusion section without any figure inside it.

- I think the literature review must be more comprehensive by adding some papers in starting sections about traditional production line, robots and cycle time to the list since these keywords are implied in the manuscript. [a] Balancing and scheduling assembly lines with human-robot collaboration tasks, COR, vol. 140, pp. 105674 [b] A framework for stochastic scheduling of two-machine robotic rework cells with in-process inspection system, CIE, vol. 112, pp. 492-502

- There are some typo errors in this manuscript. I have listed some of them. The manuscript required a full proofread.

Page 2: First, section 2 presents --> First, Section 2 presents

Page 16: requires a EnergySupply called --> requires an EnergySupply called

Comments on the Quality of English Language

Moderate editing of English language required.

Author Response

Reply to reviewer I. - # applsci-2930152

Title: Knowledge graph-based framework to support human-centered collaborative manufacturing in Industry 5.0

Authors: László Nagy, Janos Abonyi and Tamás Ruppert

Special Issue "The Future of Manufacturing and Industry 4.0"

Dear Reviewer,

We are grateful for your constructive critique and useful comments. We have addressed all the comments as explained below and highlighted them in blue in the revised manuscript. We hope that the modifications have significantly improved the quality of the manuscript.

Sincerely yours,

Tamás Ruppert 

- Some concepts required rigid definitions. For example, “Meta” is appeared in words such as meta element, meta information, meta data, metamodeling, and meta block. I wonder what is the meaning of meta here?

Thank you, we have defined these in Section 3.

- It is good to have some examples in such a framework to clarify some concepts and activities. Figure 7 is one of them and I believe it is a good Gantt chart example where human and robot spend same time on a product in Result 28 and 31. However, Result 29 and 30 are different. It would be nice if authors can further explain the meaning if the “idle time” for such a case.

Thank you for this valuable comment. We have expanded the description where we stated that the reason for the robot's idle time is that it has to wait while the operator finishes his task, since they are working at the same workstation.

- In written research papers, titles are not used on figures or tables; instead, the information is placed in a caption. Captions for tables are placed above the table (typically left aligned), and captions for figures are placed below the figure. Accordingly, authors should place captions for tables above the tables in this paper.

According to the MDPI format guidelines, table captions should be placed below the tables. We have removed the title from Figure 16 and extracted the caption.

- Figure 3 shows three different types of collaboration including Sequential (2.), simultaneous (3.) and supportive (4.) types of human-robot collaborations. However, I believe that adding (1.) Separate work to this figure can make it more comprehensive.   

Thank you, we have made the exchange.

- I think the literature review must be more comprehensive by adding some papers in starting sections about traditional production line, robots and cycle time to the list since these keywords are implied in the manuscript. [a] Balancing and scheduling assembly lines with human-robot collaboration tasks, COR, vol. 140, pp. 105674 [b] A framework for stochastic scheduling of two-machine robotic rework cells with in-process inspection system, CIE, vol. 112, pp. 492-502

We appreciate your comments and have added some relevant text and references to Section 2. We have also revised the entire section 2 and moved some relevant text here from section 3 to make the paper more comprehensive.

MINORS

- Authors should not use both British and American spelling in one article. Please do not mix the two in a single piece of writing. I have observed both of them in the paper like: utilise, utilize, behaviour, behavior, organise, organize, summarise, summarize, recognise, recognize, characterise, characterize, minimise, minimize, optimise, optimize.

Thank you very much, we have revised the manuscript and corrected it as a British version.

- Figure 16 is in the conclusion section. Please move it to the top of this section to have a conclusion section without any figure inside it.

Thank you, we put it in the right place.

- There are some typo errors in this manuscript. I have listed some of them. The manuscript required a full proofread.

Page 2: First, section 2 presents --> First, Section 2 presents

Page 16: requires a EnergySupply called --> requires an EnergySupply called

Thank you, we have corrected it.

Reviewer 2 Report

Comments and Suggestions for Authors

Dear Authors,

Thank you for submitting the manuscript.

The manuscript discusses a framework for supporting human-centered manufacturing processes in Industry 5.0. It introduces the Human-Centric Knowledge Graph (HCKG), utilizing ontologies and standards to model factors related to operators, such as movement, working conditions, and collaboration with robots. The framework aims to optimize collaboration between humans and machines in production, emphasizing the integration of human factors into the digitalized manufacturing environment. With knowledge graphs, it enables data analysis focused on improving the interaction between workers and machines. The article, through a case study on wire harness assembly, demonstrates the use of HCKG, highlighting its potential to enhance manufacturing operations management (MOM) with modules for monitoring, support, and evaluation of operator activities. This approach supports the goals of Industry 5.0, prioritizing human-centered manufacturing processes where the workforce is seen as a key, agile, and flexible resource.

The structure of the manuscript is well-conceived.

In the introductory section, the authors guide the reader into the topic, highlighting definitions of key elements of the manuscript such as MES and ERP systems, knowledge graphs, Industry 4.0, CPS, H-CPS, Resilient Operator 5.0, etc.  I believe it would be beneficial for the authors to more precisely define the term "Industry 5.0". This term has been frequently used recently, but mostly in the context of Society 5.0, not as a marker of a new industrial revolution that would be named Industry 5.0. This implies that the terms Industry 4.0 and Industry 5.0 do not represent the same level of significance.
Also, in the introductory part, it would be necessary to define the acronym KPI before starting to use it in the text.

In the second chapter of the submitted manuscript, the authors have presented a review of the state of affairs in the field of the issue at hand, with a focus on ontologies. It would be beneficial if, in this review, the authors drew from more original, primary literature sources that deal with areas of ontologies, cyber-physical production systems, HMI and HRI, and similar topics.

The main contribution of the manuscript to science is discussed by the authors in section 3. The authors present the Human-centered knowledge graph-based concept extensively and vividly through the description of provided figures and tables. It would be beneficial if the authors avoided citing literature in this section. Discussing the concept results in the loss of the originality of the authors' work due to numerous citations. I suggest that the authors move all the necessary literature for substantiating the concept to section 2 of this manuscript. This way, the authors' work will be clearly highlighted.

The application conceptual model based on Human-centered knowledge graphs is presented in the fourth section of this manuscript. The chapter is systematically and clearly structured, allowing the authors to demonstrate the implementation of the developed concept in an industrial environment and its applicability. I suggest that the authors improve the quality of the figures presented in this section.

The authors conclude the manuscript with the section Conclusions and Future Work. This section is well-conceived and provides a vivid overview of the manuscript's achievements. I suggest that the authors move Figure 16 to the fourth section of the manuscript and try to avoid citing literature in the conclusion of the manuscript if unnecessary.

Author Response

Reply to reviewer II. - # applsci-2930152

Title: Knowledge graph-based framework to support human-centered collaborative manufacturing in Industry 5.0

Authors: László Nagy, Janos Abonyi and Tamás Ruppert

Special Issue "The Future of Manufacturing and Industry 4.0"

Dear Reviewer,

We are grateful for your constructive critique and useful comments. We have addressed all the comments as explained below and highlighted them in blue in the revised manuscript. We hope that the modifications have significantly improved the quality of the manuscript.

Sincerely yours,

Tamás Ruppert 

I believe it would be beneficial for the authors to more precisely define the term "Industry 5.0". This term has been frequently used recently, but mostly in the context of Society 5.0, not as a marker of a new industrial revolution that would be named Industry 5.0. This implies that the terms Industry 4.0 and Industry 5.0 do not represent the same level of significance.

Thank you for this valuable feedback. We have expanded the description of Industry 4.0 and 5.0 in the Introduction section.

Also, in the introductory part, it would be necessary to define the acronym KPI before starting to use it in the text.

Thank you, we corrected it.

In the second chapter of the submitted manuscript, the authors have presented a review of the state of affairs in the field of the issue at hand, with a focus on ontologies. It would be beneficial if, in this review, the authors drew from more original, primary literature sources that deal with areas of ontologies, cyber-physical production systems, HMI and HRI, and similar topics.

We appreciate your comments and have added some relevant text and references to Section 2.

The main contribution of the manuscript to science is discussed by the authors in section 3. The authors present the Human-centered knowledge graph-based concept extensively and vividly through the description of provided figures and tables. It would be beneficial if the authors avoided citing literature in this section. Discussing the concept results in the loss of the originality of the authors' work due to numerous citations. I suggest that the authors move all the necessary literature for substantiating the concept to section 2 of this manuscript. This way, the authors' work will be clearly highlighted.

Thank you very much, we have moved some parts to section 2 and we have significantly reduced this section by removing some unnecessary text.

The application conceptual model based on Human-centered knowledge graphs is presented in the fourth section of this manuscript. The chapter is systematically and clearly structured, allowing the authors to demonstrate the implementation of the developed concept in an industrial environment and its applicability. I suggest that the authors improve the quality of the figures presented in this section.

Thank you for your suggestion. We have increased the size of the numbers to make them easier to read.

The authors conclude the manuscript with the section Conclusions and Future Work. This section is well-conceived and provides a vivid overview of the manuscript's achievements. I suggest that the authors move Figure 16 to the fourth section of the manuscript and try to avoid citing literature in the conclusion of the manuscript if unnecessary.

Thank you, we have put it in the right place. We have shortened the conclusion by removing the unnecessary citations and text.

Reviewer 3 Report

Comments and Suggestions for Authors

This a very interesting paper which presents the design concept of a Human-Centred Knowledge Graph. The concept is well explained and the results are well presented and implemented in the real industrial system, which is definitelly a plus. It is very useful to see a authors' view and approach to this topic and I am sure paper will be very beneficial to the journal readers. Congratulations on the great work!

However, I have a minor concern about the 57% iThenticate match, most of those are work by the authors, but please check the text and provide minor revisions to lower the percent match. 

Author Response

Reply to reviewer III. - # applsci-2930152

Title: Knowledge graph-based framework to support human-centered collaborative manufacturing in Industry 5.0

Authors: László Nagy, Janos Abonyi and Tamás Ruppert

Special Issue "The Future of Manufacturing and Industry 4.0"

Dear Reviewer,

We are grateful for your constructive critique and useful comments. We have addressed all the comments as explained below and highlighted them in blue in the revised manuscript. We hope that the modifications have significantly improved the quality of the manuscript.

Sincerely yours,

Tamás Ruppert 

This a very interesting paper which presents the design concept of a Human-Centred Knowledge Graph. The concept is well explained and the results are well presented and implemented in the real industrial system, which is definitelly a plus. It is very useful to see a authors' view and approach to this topic and I am sure paper will be very beneficial to the journal readers. Congratulations on the great work!

However, I have a minor concern about the 57% iThenticate match, most of those are work by the authors, but please check the text and provide minor revisions to lower the percent match. 

We appreciate your encouraging feedback. In response to your query about similarity, we have included some of our related results in a book chapter; however, they have not yet been published this in a journal article. We have revised the complete manuscript, removing any redundant content from prior publications.

Reviewer 4 Report

Comments and Suggestions for Authors

Research Question :

How can a knowledge graph-based framework enhance the efficiency and ergonomics of assembly line operations through the evaluation of worker movements, collaboration with machines, and in a practical case study ?

Knowledge graph pipeline :

Development of the industry-specific human-centered knowledge graph

the developed Knowledge Graph applied to a case study, divided into six groups of ontology classes. This graph includes relationships labeled on arrows and uses prefixes from other ontologies to enhance its structure, covering smart manufacturing, sensor interactions, and human-centric aspects.

Various network metrics and analytical features.

Let's break down each metric and its application:

1.       Centrality Computation

- **Purpose:** Identifies the most critical or influential objects (nodes) in the network.

- **Application:** Helps to detect significant factors in an operator's environment that influence the manufacturing process or workflow efficiency.

2.       Similarities between Nodes and Edges

- **Purpose:** Measures how similar two objects are based on their properties and their connections to other objects in the graph.

- **Application:** Addresses allocation problems by finding the best match between operators and resources, ensuring optimal resource distribution and task assignment.

3.       Flows and Paths

- **Purpose:** Determines the most efficient paths in the graph, which could be the shortest, cheapest, or quickest ways to perform a process step.

- **Application:** Optimizes the layout of the shop floor to meet operator needs more effectively, improving productivity and reducing unnecessary movements or delays.

4.       Cycles

- **Purpose:** Identifies cycles within the graph, which are sequences of nodes and edges that start and end at the same node.

- **Application:** Analyzes the distribution and repetition of tasks between humans and machines, aiming to enhance collaboration and efficiency in a work environment.

5.       Network Communities

- **Purpose:** Discovers communities within the production network, which are groups of nodes more densely connected among themselves than with the rest of the network.

- **Application:** Supports the design of human-machine collaboration by facilitating the formation of work cells or teams that can work together more effectively.

In summary, these network metrics and analytical features of KGs are used to understand and improve the manufacturing process, from optimizing resource allocation and workflow to enhancing human-machine collaboration. By analyzing the structure and relationships within the knowledge graph, decision-makers can identify areas for improvement and implement changes that lead to increased efficiency and productivity.

Limitations

Here are some possible limitations that might arise in the context of this study:

1.       Data Integration and Quality Issues

- **Heterogeneity and Inconsistency:** Integrating data from various sources into a unified KG can lead to inconsistencies and heterogeneity in data representation, making it challenging to achieve a seamless semantic integration.

For more details:

**Data Heterogeneity**

- **Semantic Heterogeneity:** Even when data formats are aligned, the meaning and context of similar data points can differ across sources, leading to misunderstandings or incorrect data mapping.

**Data Inconsistency**

- **Conflicting Information:** When integrating data from multiple sources, there may be conflicting information about the same entity or relationship, making it hard to determine which data is accurate.

- **Duplication:** Redundant data entries can occur, leading to inefficiencies and potential errors in analysis unless properly identified and merged.

- **Data Quality:** The effectiveness of KG-based analytics heavily depends on the quality of the data fed into the KG. Inaccuracies, incomplete data, or outdated information can lead to erroneous conclusions and decisions.

2.       Scalability and Performance

- **Handling Large Datasets:** As the KG grows with more detailed information and complex relationships, the system may face scalability issues, affecting query performance and response times.

- **Complex Query Processing:** Complex SPARQL queries, especially those involving extensive data or complicated relationships, can strain the system, leading to longer processing times and requiring more computational resources.

3.       Semantic Representation and Interpretation

- **Ambiguity in Data Interpretation:** The semantic representation of data in KGs might lead to ambiguity in interpretation, especially when integrating diverse data sources with different ontologies.

- **Complexity in Ontology Management:** Maintaining and updating the ontologies to accurately reflect the domain knowledge and changes in the manufacturing process can be complex and time-consuming.

Due to certain limitations, we can replace them with the use of Knowledge Graph (KG) embeddings for a study. This approach involves harnessing the detailed, structured data encapsulated within a KG and converting it into a format that machine learning models can more readily interpret. The primary goal here is to unlock the KG's potential in various applications, whether it's boosting recommendation engines, enhancing semantic search capabilities, or other objectives that leverage the KG's rich informational landscape.

Author Response

Please reject this review according to our emails.

Round 2

Reviewer 1 Report

Comments and Suggestions for Authors

I have read this manuscript once more to further check its quality.

It is improved in the revision, and so it can be accepted as it is.

Reviewer 2 Report

Comments and Suggestions for Authors

Dear Authors,

Thank you for considering the comments and suggestions for improving your manuscript. I believe that your manuscript has met the standards for publication in the scientific journal Applied Sciences.